# Biodesign and plant sciences: Evolving STEAM pedagogies in higher education

Giovanna Danies[1] , Carolina Obregón[2], Santiago Ojeda-Ramírez[3] and Andrés Burbano[4]

[1]School of Architecture and Design, Department of Design, Universidad de los Andes, Bogotá, Colombia; [2]School of Fashion, Parsons School of Design, The New School, New York, NY, USA; [3]School of Education, University of California, Irvine, CA, USA; [4]School of Arts and Humanities, Universitat Oberta de Catalunya, Barcelona, Spain

## Perspective

**Keywords:**
biodesign; biomimicry; biotech education; design thinking; sustainable futures.

**Corresponding author:**
Giovanna Danies;
Email: g-danies@uniandes.edu.co

**Associate Editor:**
Dr. Olivier Hamant

## Abstract

This perspective paper examines biodesign pedagogy in higher education, focusing on the integration of plant sciences with design and technology. We propose a dual framework for teaching biodesign: nature-driven and socially driven approaches. The nature-driven approach draws inspiration from biological strategies or biotechnologies to address environmental and societal challenges, while the socially driven approach begins with identifying societal problems and exploring biological solutions. Drawing on seven years of teaching experience, we highlight student-led projects that illustrate each approach, including eco-friendly textiles derived from plant fibres and genetically engineered crops designed for sustainable urban agriculture. Our findings underscore the potential of biodesign to bridge STEM and creative disciplines, fostering interdisciplinary collaboration, enhancing scientific literacy and equipping students to tackle complex real-world challenges.

## Introduction

Plant sciences is the study of plant growth, reproduction, evolution and adaptation to diverse environments and diseases, as well as using plants for food, medicine, fibre and ornamental purposes. Plants form most of Earth's biomass and are essential for ecosystems and human survival. Colombia's rich botanical legacy, dating back to pivotal 18th-century expeditions, combined with its position as one of the world's most biodiverse nations, makes it uniquely positioned for plant sciences (Kalin De Arroyo et al., 1994). This botanical wealth offers both ecological and cultural significance, providing sustainable solutions through plant-based biotechnologies that align with circular design principles.

Plant biotechnology and soil- and plant-associated microbes have been applied to address challenges such as climate change, social inequities and emergent public health concerns (Robinson et al., 2022). Despite this, the recent protests against genetically modified food (Anyshchenko, 2019) or against COVID-19 vaccines (Hotez, 2020) are proof of the apprehension and scepticism that some people have about biotechnology. Walker and Kafai (2021) chart the expansion of biotechnology accessibility over the last decade through community labs and toolkit development, pointing out the challenges of incorporating such complex biological processes into learning environments due to the unpredictable nature of living materials. They emphasize the need for K-12 education to evolve from traditional inductive learning to a design-oriented approach that leverages biological knowledge for creating new materials, noting the importance of initiatives like Building with Biology, which engaged the public in hands-on activities and discussions about synthetic biology through interactive kits and events (Palmer et al., 2016), and the Biodesign Challenge, a global competition that brings together students, scientists and designers to develop innovative biotechnological solutions, encouraging interdisciplinary collaboration and real-world problem-solving through project-based learning and speculative design. The Biodesign Challenge, launched in 2016, provides a platform for students from diverse backgrounds to develop biology-based solutions for environmental and social issues through hands-on projects, guided by rigorous evaluation criteria. Initially focused on undergraduate- and graduate-led initiatives, the Challenge expanded in 2019 to include high school student-led projects, broadening its reach and impact.

Additionally, universities have fostered interdisciplinarity by creating learning environments that allow disciplines such as architecture and design to interact with biological sciences. The BioArqDis Lab at Universidad de los Andes (Uniandes) in Bogotá, Colombia, for example, is a collaborative space that takes full advantage of the university's diverse academic resources across the Schools of Sciences, Engineering and Architecture and Design. The lab utilises teaching facilities during off-hours to host workshops and support students working on biodesign projects by providing space to develop prototypes and test ideas. It also makes use of specialised research labs and core facilities for specific projects, enhancing both research and educational opportunities. By connecting students with a wide range of experts and offering access to diverse spaces, the lab grounds them in the applicability of biotechnology to address challenges across multiple disciplines (BioArqDis Lab | Uniandes, n.d.). This multidisciplinary approach fosters collaboration among various stakeholders, ensuring that solutions are both innovative and practical.

Marklin Reynolds and Hancock (2010) emphasise that interdisciplinary fields like environmental biotechnology demand more than just mastery of core concepts – students must develop creative problem-solving abilities that match the complexity of real-world challenges. Spaces like the BioArqDis Lab, where expertise flows across traditional boundaries, prepare students to navigate the uncertainties of emerging fields. This experience cultivates the flexibility and collaborative mindset essential for tackling pressing environmental and technological issues.

STEAM (Science, Technology, Engineering, Arts and Mathematics) education, which integrates Arts into STEM fields, has been shown to increase participation in STEM fields (Peppler, 2013), particularly in global south contexts (Avendano-Uribe et al., 2022). When used in a *mutually beneficial and pedagogical* approach, STEAM approaches could advance both artistic and STEM fields alike (Mejias et al., 2021). Likewise, as artists and designers have had a role in challenging scientific research and its societal implications (Damm et al., 2013), the teaching and learning of biodesign, a discipline that combines design thinking methodologies with those that come from the biological sciences (Myers, 2012a), can disrupt and improve how scientific literacy and design are taught and learned (Danies Turano et al., 2020). Moreover, Walker et al. (2023) explored the impact of biodesign as a framework for making with biology (BioMaking) on K-12 life science education, highlighting its capacity to widen engagement opportunities in a culturally relevant and responsive manner. They pointed out a unique aspect of BioMaker projects, which not only allows young individuals to apply their knowledge to areas significant to them, such as personal interests and socio-political issues, but also emphasises the role of design. This engagement extends beyond fostering creativity and solving problems to include critical thinking exercises about their surroundings, underlining the educational and empowering potential these projects offer.

In this perspective paper, we describe two approaches to biodesign teaching and learning in higher education, drawing on our seven years of experience teaching and learning biodesign at a university level. Our teaching approach has consistently fostered exceptional student-led projects that have earned prestigious international recognition. Notably, our students have won the Biodesign Challenge Overall Prize twice, as well as the PETA Prize for Animal-Free Wool and the MANA Prize for the Future of Beauty. These accolades reflect the effectiveness of our pedagogical methods and highlight the power of biodesign to drive innovation while addressing complex, real-world challenges. One standout example is LixiLab, a groundbreaking soil bioremediation technology that uses the bacterium *Lysinibacillus sphaericus* CBAM5 to extract heavy metals. This project earned the Outstanding Field Research Prize at the 2021 Biodesign Challenge. Since then, LixiLab has continued to evolve, securing second place at Cumulus Green 2022 and first place at the 2024 Transformative Research Challenge, further cementing the impact of their work in the field. With the nature-driven and socially-driven biodesign approaches, we trace a framework that could be used to teach biodesign by following a project-based learning approach typical of university-level design curricula. Moreover, we will describe the affordances and benefits of each approach and argue for their use in higher education biodesign teaching. For each approach, we will illustrate our reflection with student-led projects that emerged from biodesign classes that connect plant sciences with design. We also include additional projects that are publicly available and help exemplify the connection between arts and plant science. We reflect on the potential of biodesign to disrupt both STEM and Art and Design stagnant practices in ways that science and non-science majors find in biodesign an opportunity to engage in *epistemic intersections* (Bevan, 2020) between STEM and creative fields.

## Methodology

Adopting a methodology inspired by Walker et al. (2023), we analysed project themes and student engagement within the context of the Biodesign Challenge. Specifically, we identified two coded categories – conceptual groupings developed systematically from project data and observations. These categories, informed by prior research on cultural relevance and interdisciplinary learning, provided a structured framework to uncover meaningful patterns in the ways students engaged with biodesign and addressed real-world challenges. The Biodesign Challenge served as the primary platform for the development of the projects listed in Table 1. This global educational programme brings together students, educators and professionals across design, art and biotechnology to create innovative solutions for pressing societal and environmental challenges. This structure provides participants with a robust yet flexible foundation, enabling interdisciplinary exploration while ensuring that projects remain grounded in real-world applications. Contrary to applying pre-defined categories directly, our approach began with an inductive examination led by the course instructor, focusing on the connection between biodesign projects and the everyday lives of young adults who participated in each project. This initial step was grounded in the lived experiences of the instructor and the observable impacts of these projects on students, fostering a primary layer of understanding based on direct observation and interaction.

Through this inductive process, we identified themes and patterns that reflected how projects resonated with undergraduate students' personal and academic lives. These findings highlighted the intrinsic value of biodesign in engaging students with real-world applications and challenges, thereby providing a richer, more contextualised view of learning in action. Following the inductive phase, our team connected these emergent themes to established knowledge about the benefits of STEAM learning and epistemic intersections (Bevan, 2020), for example. This step allowed us to refine our codes and deepen our analysis, situating the Biodesign Challenge as an educational activity and a conduit for meaningful learning experiences that align with broader pedagogical objectives.

**Table 1.** Biodesign projects from Universidad de los Andes in Bogotá, Colombia, which are publicly available on the Biodesign Challenge website.

| Project name | Student motivation and connection |
| --- | --- |
| RootKnit 2017 | The team was motivated by the scientific evidence they found on how our relationships with plants contribute to human health (Hall & Knuth, 2019) and the qualitative evidence they gathered during their exploration phase, which concluded that youth have little knowledge of how to take care of or communicate with their plants. By translating plant data into surface movements and shapes, RootKnit established an innovative way of understanding and connecting with plants (Figure 1). |
| WOOCOA 2018[a] | Team members were interested in addressing the fashion industry's wicked problems. PETA and Stella McCartney's Prize for animal-free wool motivated their project. Students were aware that Colombia has a diversity of plant-derived fibres and social justice problems that could benefit from their solution (Sullivan, 2018). Meet Woocoa. |
| YUNDO 2018 | Team members were interested in offering an alternative to improving health by enhancing indoor air quality. Yundo is a living piece of art made from agricultural waste. It contains cyanobacteria that naturally purify air in the home. The cyanobacteria turn $CO_2$ into oxygen and are engineered to bioluminescent while releasing a pleasant scent. Yundo may be applied as a piece of art, a wall or a panel in each home. |
| Alga Viva 2019 | Team members were interested in addressing $CO_2$ emissions from the transport sector. They were inspired by Bogotá's public transport 'circulatory system' called *transmilenio*. Algaviva is a microalgae urban structure designed to purify the air surrounding the local bus stations in Bogotá. |
| PseudoFreeze 2019[b] | Team members were inspired by the fact that a protein could immediately freeze cool water. They then searched for a context in Colombia that could benefit from this biotechnology and identified coolers for the transport of vaccines to rural areas as an opportunity. Team PseudoFreeze engineered a refrigeration system to transport vaccines, which harnesses energy from the INA protein from the bacterium plant pathogen *Pseudomonas syringae*. It requires no batteries or outside power sources. Today, it has become a start-up named NanoFreeze (n.d.). |
| SauColors 2019 | Team members' interest in addressing the fashion industry's wicked problems. ORTA Prize for Bioinspired Textiles Processes motivated the project. The team was particularly interested in addressing the indigo problem. They were actively thinking of alternatives to obtain blue pigment in nature. Walking across the university's campus, they came across blue blackbird faeces. They learned that the blackbird's digestive system employs an alkaline compound that reacts with elderberries, creating a blue colour. They replicated this process in the lab and obtained different colours across the Pantone spectrum (https://atlasofthefuture.org/project/saucolors/). |
| Growing Color 2019 | Team members were interested in addressing the fashion industry's wicked problems. The ORTA Prize for Bioinspired Textile Processes motivated the project. Growing colour involved genetically manipulating cotton plants to eliminate textile dying processes. |
| Filling Green 2019 | Team members' interest in addressing the fashion industry's wicked problems. ORTA Prize for Bioinspired Textiles Processes motivated the project. Filling Green proposes a plant-based alternative for down feathers and polyester, using a blend of fibres including corn silk (*Zea mays*), pineapple leaves (*Ananas comosus*) and tururi sacs (*Manicaria saccifera*). This team tested the heat-retaining capacity of a material derived from agricultural waste versus conventional ones, demonstrating a better overall performance. |
| Agriculture of the Future 2019 | Team members are interested in addressing food scarcity in urban areas. This project selected potatoes as a staple food in Colombia. They propose a genetically engineered 'supercrop' nutritionally richer and easier to grow in controlled indoor environments. They propose special farming towers with a circular resource management system to produce and commercialise the super papa. |
| Papacha 2019 | The team members wanted to develop an accessible and environmentally friendly alternative to women's sanitary pads. These were made from potato scraps (an abundant crop in Colombia). |
| Scoby Bubble Wrap 2019 | Team members are interested in offering alternatives to single-use plastics. This project aims to create an eco-friendly alternative to plastic bubble wrap: a symbiotic colony of bacteria in yeast, bacterial cellulose and tea as a subtract. |
| Linneo, Nourishing Beauty[c] | Team members investigated innovative, sustainable alternatives to traditional beauty products and practices. Linneo sources cosmetic pigments from fungi, aiming to replace harmful ingredients like heavy metals, parabens and phthalates with natural, bio-based options that nourish and protect the skin. |
| LixiLab 2021[d] | Team members were interested in working with the farming community from Mochuelo Alto, Colombia, located in Bogotá's largest landfill. As pollution increases, small farming communities become more vulnerable to toxic soil quality. LixiLab proposes an accessible soil bioremediation technology to extract heavy metals using a bacterium called *Lysinibacillus sphaericus* CBAM5. The solution was co-created with the community to guarantee easy adoption of the technology. |
| Geoenergy 2022 | Students were inspired by the capability of the soil bacteria of the genus *Geobacter* to produce energy. Geonergy is a modular device designed to control frost in crops. It is powered by bacteria of the genus *Geobacter* and consists of a temperature and humidity sensor and electric heater. |
| Mus(T)Go 2022[e] | The team was motivated by the Paramo ecosystem in Colombia and searched within this ecosystem for inspiration for their project. The team became fascinated with the moss and its capability to retain water. They first thought of working on an alternative to the super-absorbent polymers found in diapers by employing proteins found in the moss. Then, they found a research paper showing the moss's capability to retain microplastics. They researched the microplastic challenge and found that, on average, we consume the equivalent of one credit card per week of plastic. The team designed a moss filter that removes harmful microorganisms and microplastics from water. |

[a] PETA Prize for Animal-Free Wool.
[b] 2019 Biodesign Challenge Overall Prize; Today a start-up named NanoFreeze.
[c] MANA Prize for the Future of Beauty.
[d] Outstanding Field Research Prize at the 2021 Biodesign Challenge; Cumulus Green 2022; First place at the 2024 Transformative Research Challenge.
[e] 2022 Biodesign Challenge Overall Prize.

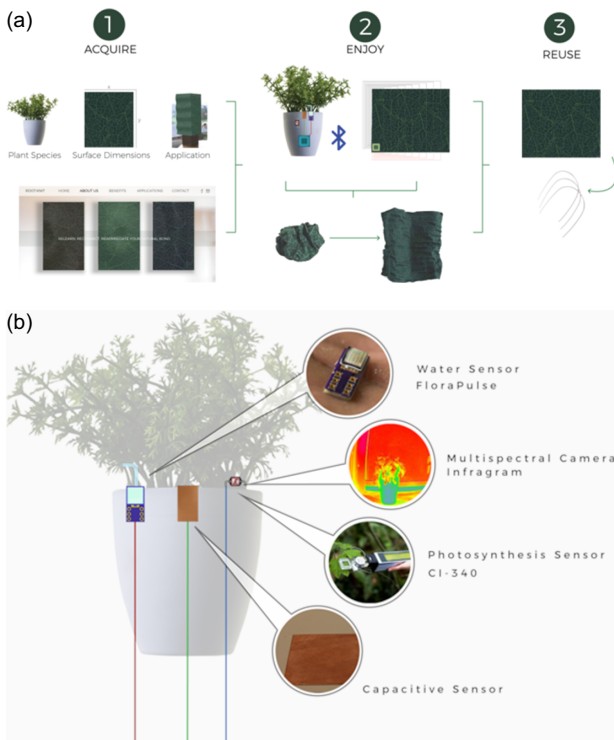

**Figure 1.** *RootKnit* (Molina et al., 2017). (a) Shape-shifting surface that reacts to the plants' conditions and then translates that data into surface movements and shapes, (b) connected to a water FloraPulse sensor, multispectral inframgram camera, photosynthesis CL-340 sensor and capacitive sensor.

Thus, our methodology diverges from a straightforward application of pre-defined coding to a more dynamic and iterative process of theme development and theoretical integration. This approach reflects a commitment to capturing the complexity and richness of how biodesign projects influence and are shaped by the participants, their learning environments and the societal and cultural contexts in which they are embedded. By focusing on higher education settings, our study contributes to a deeper understanding of the pedagogical potentials of biodesign in fostering innovation, engagement and learning beyond the pre-collegiate level.

In this paper, we revise publicly available projects that emerged from BioArqDis Lab, an interdisciplinary space that joins architecture, biology and design. The BioArqDis Lab at Uniandes serves as a physical research and innovation space that fosters interdisciplinary collaboration. The lab supports the exploration of solutions derived from biological systems or processes, focusing on biomaterials, biofashion, biotechnology and synthetic biology. Through workshops, courses and mentoring, the lab provides an environment where students and researchers can design and execute biodesign projects that connect biology, technology and creative practices. We also reflect on projects that inspire our students in this interdisciplinary thinking and designing approach.

## Results

### A framework for biodesign teaching and learning in higher education: finding opportunities for biodesign in nature or in societal problems

Analysis of student biodesign projects revealed that all successful projects followed one of two distinct approaches: nature-driven or socially driven biodesign. Nature-driven projects begin by identifying biological strategies that can address societal needs, while socially driven projects start with identifying a problem and then searching for biological solutions. This is reflected in the consistent success of our student-led projects, many of which have been recognised with prestigious awards. For instance, MUS(T)GO received the Biodesign Challenge Overall Prize in 2022, while PseudoFreeze (now known as NanoFreeze) has grown into a successful startup after winning recognition for its innovative vaccine refrigeration solution. These accolades highlight the impact of our approach in fostering excellence and addressing pressing societal and environmental issues. One approach involves directly engaging with nature to discover biological strategies or biotechnologies that can be harnessed to address social or environmental issues. This method uses nature's inherent capabilities as opportunities, where the design process revolves around finding practical applications for these natural features. For example, the discovery of ice-nucleating proteins in *Pseudomonas syringae* (Cochet & Widehem, 2000) has been adapted to improve vaccine refrigeration logistics, showcasing how natural phenomena can translate into solutions for pressing health challenges.

Conversely, the second approach starts with clearly observing a social or environmental problem – viewing these issues as opportunities for innovation. Here, the design process entails searching within nature to find solutions to address the identified challenges. This methodology ensures that biodesign solutions are deeply rooted in actual needs, enhancing the relevance and impact of the designs. Projects like SauColors and Filling Green (see Table 1) exemplify socially driven biodesign by addressing industrial environmental challenges through sustainable solutions, such as using natural dyes and plant-based fibres to reduce reliance on traditional, resource-intensive methods.

Both approaches offer insights for designing biodesign units or pedagogies. The first approach fosters a deep engagement with natural phenomena and biotechnologies, encouraging students to explore and be inspired by the intricacies of biological systems. This nurtures creativity and builds a profound appreciation and understanding of biological principles, which can be pivotal in developing innovative and sustainable design solutions. The second approach emphasises a problem-first methodology that engages students in identifying and understanding significant societal challenges before seeking biological solutions. This method encourages a pragmatic, user-centred design process highly responsive to specific community needs or environmental issues. These two approaches highlight the versatility and adaptiveness of biodesign education. We underscore the importance of a dual-focused pedagogical framework that begins with nature's inspirations or concrete societal challenges, each leading to meaningful and contextually relevant biodesign outcomes. We posit that this conceptualisation not only enriches the educational experience but also ensures that the solutions developed are informed by a nuanced understanding of the interconnectedness of biological, social and ethical dimensions.

### Nature-driven biodesign: opportunities for biodesign in nature

Living organisms have developed strategies to navigate challenges such as solar radiation, extreme temperatures and water scarcity (Innovation Inspired by Nature – AskNature. (n.d.), Lindgren et al., 2016). In particular, plant sciences stand out as a crucial foundation for biodesign, given plants' dominance in global biomass and their role as a culturally and ecologically significant resource, especially in biodiverse regions like Colombia (Danies Turano et al., 2020).

These biological adaptations provide a rich source of inspiration for innovative solutions in biodesign. The Biodesign Challenge has revealed several inspiring projects that demonstrate the efficacy of biodesign inspired by nature. Regarding the projects we have analysed, for instance, NanoFreeze (n.d.) utilised ice-nucleating proteins sourced from the plant pathogen *Pseudomonas syringae*, which naturally form ice crystals within plant tissues. This biotechnology has been ingeniously adapted to improve the efficiency of refrigerating vaccines during their transportation to remote areas in Colombia, showcasing how biodesign can ingeniously solve critical logistical and public health challenges. Another impactful project is MUS(T)GO, which emerged as the 2022 overall winners of the biodesign challenge. This initiative began with a detailed exploration of the páramo ecosystems in Colombia's Chingaza National Park, a crucial water source for Bogotá. The team's discovery of mosses' superabsorbent and microplastic-retaining properties led to the development of a novel water filtration system designed to serve underserved communities on Colombia's Atlantic Coast. By utilizing natural moss encased in a stainless-steel helix, this project not only addresses the pressing issue of water purity but also integrates seamlessly into existing infrastructure, demonstrating a sustainable and locally adapted solution.

These case studies illustrate an approach to biodesign learning in which students engage deeply with natural phenomena – from the properties of *Pseudomonas syringae's* proteins to the unique capabilities of mosses – and creatively apply these insights to design solutions that are both innovative and environmentally sensitive. This approach suggests initiating the teaching and learning of biodesign by fostering environments and scenarios where students actively engage with and draw inspiration from nature. For instance, through field trips to diverse ecosystems, interactive laboratory experiments mimicking natural processes or reflective sessions discussing the ingenious adaptations of organisms. We posit that these experiences not only inspire creativity but also cultivate a deeper understanding and appreciation of biological principles, setting the stage for innovative and sustainable design solutions. This approach highlights the importance of nurturing a deep curiosity about natural systems and leveraging their intrinsic properties to address relevant environmental and societal challenges, thus fostering a robust and holistic educational experience in the field of biodesign.

Reflecting not only on the lived experience of a professor in a biodesign course but also on the histories behind the projects showcased in this article, we have observed that this approach of seeking biodesign opportunities in nature offers students in creative fields like design and art a chance to engage with STEM practices. These practices include planning and conducting investigations in the natural world or arguing based on evidence (Bevan et al., 2019). Additionally, through this approach, creative students contribute by designing interrelations across various sign and media languages and systems, negotiating what constitutes a 'good' project and evaluating the success of their goals from an artistic, rather than scientific, perspective. We believe that the integration of these practices enriches the experiences of STEM students, allowing them to question the purpose of science, assign new meanings and contextualise scientific practices.

## Socially driven biodesign: opportunities for biodesign in social problems

Biological solutions developed by plant scientists offer versatile templates for solving practical problems across various fields, including fashion, textiles, energy and agriculture. Leveraging design thinking, plant breeders, pathologists and biologists apply their deep understanding of plant systems to address global challenges such as increasing food nutrition, improving crop efficiency and mitigating the impacts of climate change and deforestation. This interdisciplinary collaboration between plant scientists and designers nurtures innovations that meet the needs of a growing population while conserving natural resources (Danies Turano et al., 2020).

In the context of teaching biodesign, students are often stimulated by either externally imposed challenges or by intrinsic motivation to address problems that are personally or socially relevant to their communities. This educational strategy emphasises designing with users or communities, ensuring that the solutions developed are practically implementable. The design thinking process that students undertake includes considerations of the system's overall framework, potential risks, the life cycle of products and the sustainability of solutions over their short-, medium- and long-term existence. The biodesign journey begins with students conducting in-depth, human-centred research within their own contexts, followed by an exploration of existing trends in biotechnology or related scientific fields that could further aid in addressing the problems identified (Danies Turano et al., 2020).

From the Biodesign Challenge, several projects have exemplified how this approach can be applied to environmental challenges. Sau-Colors, for instance, created a sustainable dyeing method using the anthocyanin from elderberries, producing a range of environmentally friendly colours and offering an alternative to synthetic dyes by manipulating variables such as pH, temperature and mordants (Echeverry et al., 2019; Figure 2). Similarly, Filling Green developed a promising fibre blend to replace down and polyester in insulation materials based on the research team's quantitative results. Three key properties – thermality, softness and compression resilience – guided the material-selection process. Thermal tests on individual fibres, which are byproducts of agricultural processes, – tururi sack, pineapple leaf fibre and corn silk – demonstrated strong insulation performance comparable to or better than down. The uninsulated sample exhibited the fastest temperature drop, underscoring the importance of insulation. Among the fibres tested, corn silk showed the slowest heat loss, outperforming even down, while pineapple leaf fibre and tururi sack followed closely, with thermal curves similar to or better than down. The team then created a blend consisting of 45% tururi sack (for density), 20% pineapple leaf fibre and 35% corn silk (for softness and thermal performance). This mixture, the Filling Green Cocktail, consistently demonstrated lower heat loss than down over the 12-minute test period, confirming superior insulation. Additionally, antimicrobial tests under extreme conditions – including steam, cold and enclosed storage – showed no deterioration in odour, texture or integrity, emphasising the material's durability. The team recommended felting as the manufacturing method to ensure consistency and volume, offering a sustainable, high-performance alternative to synthetic and animal-derived fillers for cold-weather apparel (Cardona et al., 2019; Figure 3).

Some projects focused on personally relevant themes or health-related issues, connecting deeply with the students' own experiences or the needs of their communities. The project YUNDO, for example, conceptually addresses the issue of indoor air pollution using cyanobacteria. This live art wallpaper not only purifies air but also enhances indoor environments aesthetically and sensorially, adding bioluminescence and pleasant scents to improve the quality of indoor living (Gaitán et al., 2018; González-Martín et al., 2021).

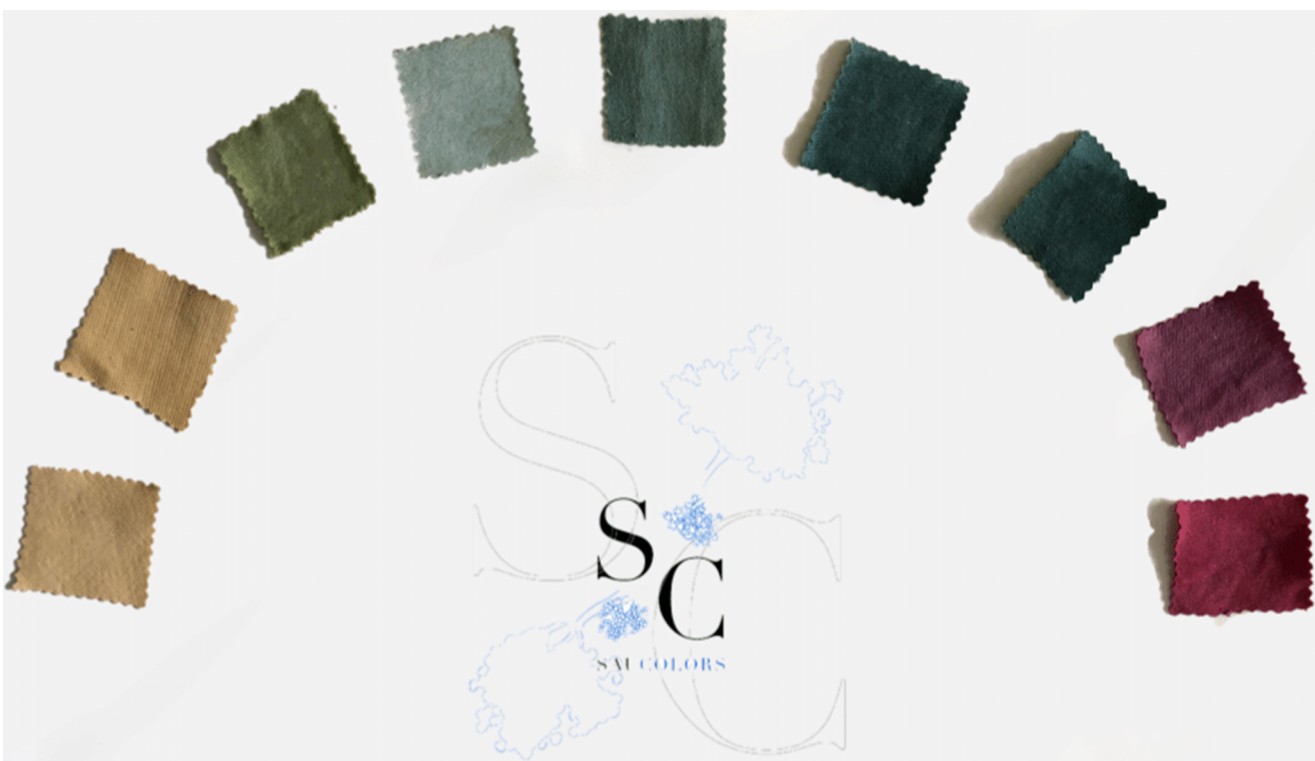

**Figure 2.** *SauColors* (Echeverry et al., 2019). Natural dyes derived from the anthocyanin content found in elderberries fruits produced on the *Sambucus nigra* tree. When mixed with an alkaline solution, under specific pH, temperature, time and mordant, a range of colours that range from blue and purple to beige and green can be generated.

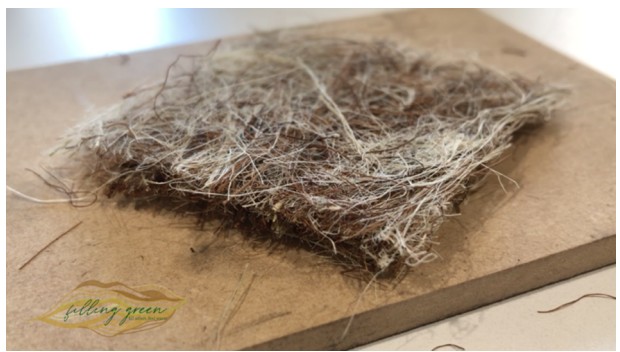

**Figure 3.** *Filling Green* (Cardona et al., 2019). A plant-based alternative to down and polyester feathers, using a blend of bioproducts from corn silk (*Zea mays*), pineapple leaves (*Ananas comosus*) and tururi sacs (*Manicaria saccifera*).

This problem-centred approach to teaching biodesign encourages students to start with a thorough understanding of significant societal or personal issues and then explore biological solutions. It prompts an educational pathway where students are engaged in scenario-based learning, simulating or enacting real-life problems, and employing reflective practices to critically evaluate the societal and environmental implications of their designs. Starting with a problem-first methodology not only helps students develop a deeper understanding of the issues at hand but also fosters a more meaningful application of biodesign solutions, leading to innovative and contextually appropriate outcomes. This approach underlines the potential of biodesign education to not just respond to but anticipate the needs and challenges of the future, ensuring that students are well equipped to apply their knowledge in real-world contexts.

This approach also allows for an interweaving of scientific, artistic and creative practices. Finding purpose is essential for powerful learning (Hilton & Pellegrino, 2012), and we believe that by finding opportunities for biodesign in societal problems, students from STEM disciplines find purpose for their scientific background, discovering usability and meaning. After reflecting on the projects that emerged through this approach, we posit that finding opportunities for biodesign in existing problems allows STEM students to engage in epistemic intersections between arts and science practices, such as the meaning-making practices described by Bevan et al. (2020). These include finding relevance, considering multiple approaches and adopting a critical stance when facing problems. In other words, through this approach, STEM students 'fall in love' with a problem, enabling them to renegotiate the purpose and meaning of scientific knowledge.

## Discussion

The two approaches we identified can guide educational researchers in describing the practices students learn through biodesign. Our reflections suggest that a key skill students acquire while learning biodesign is identifying and addressing design opportunities in nature or societal problems. In traditional scientific learning, thinking processes focus on testing hypotheses and building theories. Biodesign, however, encourages students to consider questions like 'what if' plant pathogenic mechanisms could be leveraged in other contexts to solve environmental or social problems, for example: What if plant defence strategies against pathogens are applied to societal challenges? Studying the plant pathogen *Phytophthora infestans* traditionally seeks cause–effect explanations, whereas biodesign views disciplinary knowledge as an opportunity to address real-life issues, applying scientific knowledge to

design. Whereas we found that students are either engaged in finding opportunities for biodesign in nature or opportunities for biodesign in social problems, our findings also show that biodesign learning experiences can still engage with more depth in artistic processes. Both nature-driven and socially driven biodesign are focused on applied solutions, but emerging artistic domains such as BioArt depart from being solution-centred and foregrounds active questioning and reflecting on what art can be and around what material we can use while making art (Mitchell, 2015).

We think a BioArt approach to biodesign education can be enriching for several reasons. BioArt immerses students deeply into biotechnology, mirroring the creative processes of artists and designers. This approach critically examines technologies typically not associated with art, combining biotechnology and art to explore new fields innovatively. For instance, it leads to breakthroughs in biofashion and other domains where production and creative processes were previously limited. Biotechnologies now enable artists and designers to collaborate with biologists and scientists, continually seeking new mediums and advancing their work. Through BioArt approaches to biodesign education, instructors and students can tackle socially and personally relevant problems deeply. This aligns with the notion that art raises questions (Dewey, 2008) while design seeks solutions (Buchanan, 2001), breaking down traditional boundaries between art and science, nature and culture and the real versus artificial. This expanded perspective is essential for addressing complex systemic issues, exemplified by artworks like Eduardo Kac's 'Edunia' (Kac, 2007) and Gilberto Esparza's 'Autophotosynthetic Plants'. Such examples could inspire biodesign education approaches where students formulate questions reflecting the interconnectedness of biological systems and societal needs, merging creativity with scientific inquiry to ensure biodesign solutions are informed by a nuanced understanding of biological, social and ethical dimensions.

## Conclusions

In this perspective paper, we have reflected on existing projects that utilised plant science to fuel and inspire biodesign initiatives. Moreover, we examined how students engaged with each project and discovered that students typically followed one of two paths. They either identified opportunities in nature and then sought socially relevant contexts to apply those natural features or biotechnologies for societal benefit, or they found opportunities in existing social problems and then explored available biotechnologies to address or advance solutions to those problems.

The integration of scientific knowledge with creative methodologies that engage communities empathetically can empower students to develop ethically responsible solutions, minimizing negative impacts on the environment and communities. Such projects can also enable students to pursue fulfilling careers capable of addressing unique challenges. Teaching biodesign does not require instructors to be primary experts in biotechnologies and design. Instead, they can guide students in contextualizing biotechnology and connect them with experts for deeper learning (Jimenez et al., 2022). Moreover, the centrality of plant sciences to biodesign stems not only from their sustainability potential but also from their cultural and ecological relevance, particularly in biodiverse contexts like Colombia, where they offer both inspiration and practical value for innovative solutions. The science and design pedagogy can discuss biotechnology in depth while encouraging students to find solutions encompassing human, social and cultural practices.

We propose a framework for biodesign learning that involves either identifying design opportunities in social contexts and applying existing biotechnological knowledge or repurposing fascinating natural features or biotechnologies for social contexts. This framework can guide both researchers and biodesign educators. It suggests tailoring biodesign lessons to the disciplinary background of students: for STEM students, the focus could be on understanding social contexts and identifying relevant biotechnologies; for students from creative fields like design or arts, the focus could be on deeply understanding a biotechnology and then finding a problem that it can solve.

**Open peer review.** To view the open peer review materials for this article, please visit http://doi.org/10.1017/qpb.2025.10022.

## Acknowledgments

We thank the Biodesign Challenge for encouraging the creation of global education programmes at the intersection of design, art and biotechnology, as well as the School of Architecture and Design at Universidad de los Andes for their growing support of the biodesign program.

**Competing interests.** The authors declare none.

**Disclosure statement.** There are no competing interests to declare. This manuscript has not been published nor is it under consideration by any other journal. There are no financial obligations with governmental or private organisations that can affect the content, results or conclusions of the present manuscript.

**Author contributions.** GD and SO conceptualised and designed the study. Data collection was carried out by GD, CO and AB. The analyses were performed by GD and SO. The manuscript was written collaboratively by GD, SO, CO and AB, with all authors contributing to the drafting, revision and final approval.

**Funding statement.** This research received no specific grant from any funding agency, commercial or not-for-profit sectors.

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
