## [Reviewer Report]

Thoroughly researched and evidence-based paper. Appreciate the succinct way the authors define the two current pathways into Biodesign. Solid case studies of their students' projects to illustrate these pathways.

One small edit: there isn’t uniformity between Biodesign, BioDesign, and biodesign. This might be intentional on the authors' part - if so, there may need to be a small section distinguishing the three spellings.

---

## [Reviewer Report]

This perspective piece describes the authors’ pedogeological learning and observations from their experience supervising student-led biotechnology products. The students are from the arts and design backgrounds. The manuscript details two general directions in which the students tend to develop their project ideas (challenge-led or biology/nature-led). Using the examples from the authors’ students, they explain how each direction could work in retrospection. This manuscript is clearly written, with appealing visuals, and provides unique insights into how collaboration with art and design could aid innovation and education in plant science. Below, I suggest several revision points for further improvement.

Major points:

1) The structure is such that the manuscript feels repetitive (two courses of ideation are mentioned many times in parallel; you could consider discussing each in depth instead.

2) Currently, it is not so clear there are two major ways students develop their projects (it sounds like these fit a few of them). Consider specifying which was employed in the past project examples in Table 1. Is one way more popularly used than the other?

3) The authors’ teams have had exceptional success developing projects (consistently high quality recognized by awards), which should be emphasized in the text. They have a brilliant track record in this space, which is why we want to read and learn about their pedagogical philosophy and methods.

4) Biodesign Challenge was the platform for which most (all?) projects in Table 1 were developed. It is also mentioned that they offer a pedagogical framework. Can you explain this program so readers can appreciate the boundaries and freedom placed on the projects?

5) The figures are fine, but it may be more helpful if the projects more majorly mentioned in the text are featured (e.g., PseudoFreeze).

6) Plant science relevance—Plant science is disproportionately represented in past projects as the biological basis and inspiration. Why is plant science particularly relevant to biodesign? This point is implied at the very beginning of the manuscript, but it is worth expanding on. For example, is it because plant-driven or plant-based biotechnology tends to be more sustainable?

Minor points:

1) More references are needed in some places, e.g., Line 32-34: “Plant biotechnology and soil…. concerns.”; Line 45-47: “thus grounding students… participate”; and Line 60-62 “BioMaker projects…. role of design.”

2) Define specific terms/programs/projects, such as the Biodesign Challenge, BioMaker Project, and BioArcDisLab. They are explained, but it is not entirely clear for a complete novice to understand them.

3) Line 80-81: “three coded categories” – what are they?

4) Table 1: “How is this project connected to youth and their everyday life? What motivated youth in this project?” – can this be summarized, e.g., “student motivation and connection”?

---

## [Editor Report]

Dear authors,

As you can see the reviewers are positive about your manuscript, and praise the original angle. Reviewer 2 has several insightful suggestions for improvements, which I invite you to consider. Regarding one of their points about the plant focus (apart from the journal name), one factor you might consider is socio-ecological robustness (when compared to other source of inspiration: 82% of Earth biomass is plant-based). 

An additional point to consider is the quasi absence of quantitative statements in the text. It would be good to mention at least a few indicators. For instance, line 214:

“Similarly, Filling Green designed eco-friendly insulative materials from plant-based fibers such as corn silk, pineapple leaves, and tururi sacs. These fibers, which are byproducts of agricultural processes, were rigorously tested for their thermal and antimicrobial properties, positioning them as viable alternatives to down and polyester in apparel (Cardona et al., 2019; Figure 3).” You could add a few indicators of thermal and antimicrobial enhanced property assessment (from the original research) to strengthen this statement. This applies to other statements too (of course, you cannot be exhaustive in this perspective, but some quantitative assessment would make your conclusions stronger)

---

## [Editor Report]

Sorry for the very long delay in our response. We had issues finding new reviewers for the revision. As the new version properly addresses the remaining (minor) points from reviewer 2, I’m happy to accept this revision.